# Comparison of the Glycolytic and Alcoholic Fermentation Pathways of *Hanseniaspora vineae* with *Saccharomyces cerevisiae* Wine Yeasts

**María José Valera [1]**, **Eduardo Boido [1]**, **Eduardo Dellacassa [2]** and **Francisco Carrau [1,*]**

[1]  Área de Enología y Biotecnología de Fermentaciones, Facultad de Química, Universidad de la República, 11800 Montevideo, Uruguay; mariajose_valera_martinez@hotmail.com (M.J.V.); eboido@fq.edu.uy (E.B.)

[2]  Laboratorio de Biotecnología de Aromas, Facultad de Química, Universidad de la República, 11800 Montevideo, Uruguay; edellac@fq.edu.uy

[*]  Correspondence: fcarrau@fq.edu.uy

**Abstract:** *Hanseniaspora* species can be isolated from grapes and grape musts, but after the initiation of spontaneous fermentation, they are displaced by *Saccharomyces cerevisiae*. *Hanseniaspora vineae* is particularly valuable since this species improves the flavour of wines and has an increased capacity to ferment relative to other apiculate yeasts. Genomic, transcriptomic, and metabolomic studies in *H. vineae* have enhanced our understanding of its potential utility within the wine industry. Here, we compared gene sequences of 12 glycolytic and fermentation pathway enzymes from five sequenced *Hanseniaspora* species and *S. cerevisiae* with the corresponding enzymes encoded within the two sequenced *H. vineae* genomes. Increased levels of protein similarity were observed for enzymes of *H. vineae* and *S. cerevisiae*, relative to the remaining *Hanseniaspora* species. Key differences between *H. vineae* and *H. uvarum* pyruvate kinase enzymes might explain observed differences in fermentative capacity. Further, the presence of eight putative alcohol dehydrogenases, invertase activity, and sulfite tolerance are distinctive characteristics of *H. vineae*, compared to other *Hanseniaspora* species. The definition of two clear technological groups within the *Hanseniaspora* genus is discussed within the slow and fast evolution concept framework previously discovered in these apiculate yeasts.

**Keywords:** glycolysis; yeast; pyruvate kinase; non-*Saccharomyces*; fermentation evolution clade

## 1. Introduction

One of the main characteristics of yeast affecting their oenological use is their capacity to ferment sugars. Non-*Saccharomyces* yeasts have traditionally been considered bad fermenters. For that reason, selected strains of *S. cerevisiae* have been used in the oenological industry to ensure that complete fermentation occurs [1,2]. Recently, however, wineries have been encouraged to apply new, non-*Saccharomyces* species in winemaking processes to provide distinguishable flavours within wines [3–6]. Non-*Saccharomyces* species have been used to produce different aromas and flavours, compared with *Saccharomyces* strains [7,8]. Therefore, many efforts have been made to identify non-conventional yeast strains for oenological purposes [4,8,9].

The selection of oenological yeasts is commonly accomplished by identifying species from raw material. Spontaneously fermented grape musts are the niche that is most commonly used to identify novel strains that are both capable of fermenting sugars and confer desirable flavours to wines [10,11]. *Hanseniaspora* is the most abundant genus on grapes and grape juices. Studies have shown that up to 75% of the yeast population during the early stages of fermentation is made up of *Hanseniaspora* species [12,13]. After the first 48–72 h of spontaneous fermentation, the percentage of

*Hanseniaspora* species present decreases and the *S. cerevisiae* strains correspondingly increase. However, some *Hanseniaspora* species have been detected throughout the fermentative process [14].

Researchers have maintained that observed changes in yeast populations during fermentation occur, at least in part, because *Hanseniaspora* species are sensitive to ethanol [15]. *S. cerevisiae* is able to produce high quantities of ethanol rapidly. Therefore, this species dominates the fermentation process until sugar is completely depleted. Recent studies have shown the effects of antimicrobial peptides secreted by *S. cerevisiae* throughout the fermentation [16,17] inhibiting the growth of non-*Saccharomyces* yeasts. Therefore, the reputation of *Hanseniaspora* species as poor fermenters may be due to the presence of other inhibitors and not directly related to their reduced capacity to ferment sugars.

Not all the *Hanseniaspora* strains have the same properties. Species of the genus produce different secondary metabolites and exhibit different fermentative behaviours. In fact, differences could even be detected between strains belonging to the same species [18,19]. *H. vineae* is an epiphyte yeast that is not easily isolated from fruit, a feature it shares with all the *S. cerevisiae* strains [20]. *H. vineae* can be isolated from samples after one or two days of spontaneous fermentation in wines and other fruit beverages such as cider [19]. This highlights the distinct behaviour of the species, compared with the majority of *Hanseniaspora* apiculate yeasts, which are commonly isolated from the skin of grapes or grapevine soil. The ability of strains identified as *H. vineae* to complete grape juice fermentation has been demonstrated via single inoculation [7]. Moreover, selected strains of *H. vineae* contribute positively to wine aromas by providing floral and honey notes, even when sequential inoculation with *S. cerevisiae* strains was performed [21]. Our assessment of *H. vineae* showed that levels of phenylpropanoid flavour compounds synthetized from grape must were elevated compared with other yeasts. In *H. vineae*, the presence of metabolic pathways that actively transform aromatic amino acids explains the elevated phenyl acetate ester and benzenoid derived compounds synthesis compared to other yeasts and these flavour compounds provide fruity and flowery aromas [21–23].

Although several phenotypic studies have been carried out throughout wine fermentation using non-*Saccharomyces* species, there is a lack of information regarding the genetic basis of observed characteristics in non-*Saccharomyces* strains [8]. Due to the development of next generation sequencing, genomes of *Hanseniaspora* species from wine have been recently sequenced [24–27]. Further work will be needed to determine which genes are responsible for each function. In previous studies, the aromatic profile of *H. vineae* was correlated with genomics and transcriptomics data [22]. However, genes involved in glycolysis and fermentative behaviour in the species remain unknown. In *S. cerevisiae*, genes necessary for fermentation have been reported using mutant analysis. All of these, were grouped in a "fermentome" [28]. The genome of *H. guilliermondii* has been recently analysed [27] and the presence and absence of genes involved in the glycolytic and fermentative pathways compared with *S. cerevisiae* and other *Hanseniaspora* species were reported. Moreover, the *H. uvarum* glycolytic pathway has been assessed in a study that revealed the catalytic potentials of enzymes involved in the route [29]. The authors showed that the main glycolytic enzyme of *H. uvarum*, pyruvate kinase, had a 15-fold lower enzymatic activity than that of the *S. cerevisiae* enzyme.

The aim of this work is to establish the differences and similarities between *H. vineae*, other *Hanseniaspora* species, and *S. cerevisiae* regarding glycolytic and fermentative behaviour. In the present study, a comparative analysis of the fermentative capacity of *H. vineae* was performed using genetic and transcriptomic data. Characterization of the glycolytic and fermentative potential of *H. vineae* will enhance our understanding about the mechanisms and the regulation of the fermentative process in a non-*Saccharomyces* yeast. *Hanseniaspora* genus studies might help reveal new signs of *S. cerevisiae* domestication mechanisms for wine production.

## 2. Materials and Methods

### 2.1. Yeast Strains

Yeast strains used for this study are listed in Table 1.

**Table 1.** Yeast strains used in this study.

| Species | Strain | Source | Use |
|---|---|---|---|
| *H. vineae* | T02/19AF | Fermenting Tannat grape must (Uruguay) | Genomic, transcriptomic, phenotypic analysis |
| *H. vineae* | T02/05AF | Fermenting Tannat grape must (Uruguay) | Genomic and phenotypic analysis |
| *H. osmophila* | AWRI3579 | Fermenting Chardonnay grape must (Australia) | Genomic and phenotypic analysis |
| *H. uvarum* | AWRI1280 | Fermenting Tannat grape must (Uruguay) | Phenotypic analysis |
| *H. uvarum* | AWRI3580 | Fermenting Chardonnay grape (Australia) | Genomic analysis |
| *H. opuntiae* | AWRI3578 | Fermenting Chardonnay grape (Australia) | Genomic analysis |
| *H. valbyensis* | NRRL Y-1626 | Soil (Denmark) | Genomic analysis |
| *H. guilliermondii* | UTAD222 | Grape must (Portugal) | Genomic analysis |
| *S. cerevisiae* | 288Sc | Laboratory strain | Genomic analysis |
| *S. cerevisiae* | ALG804 | Oenological yeast (Oenobrands®) | Phenotypic analysis |

*2.2. Fermentation in Natural Grape Must*

Chardonnay grape must containing 300 mg N/L and 200 g/L of sugars at pH 3.5 was treated with 200 mg/L dimethyldicarbonate to prevent microorganism growth. Pre-cultures of *H. vineae* T02/19AF, *H. vineae* T02/05AF, *H. uvarum* AWRI1280, *H. osmophila* AWRI3579, and *S. cerevisiae* ALG804 were isolated from the Chardonnay grape must and incubated at 25 °C for 12 h in a rotary shaker at 150 rpm. Then, 125-mL Erlenmeyer flasks closed with cotton plugs used to simulate microaerobic conditions were inoculated with 75 mL of must containing $1 \times 10^5$ cells/mL. Static batch fermentations were conducted at 20 °C to simulate winemaking conditions.

*2.3. Growth Kinetics in Different Types of Media*

Six types of growth media were prepared using yeast nitrogen base (YNB) (Difco, Detroit, MI, USA) as a sole nitrogen source (6.7 g/L). Media were supplemented with the following carbon sources: Glucose, fructose, sucrose, xylose, glycerol, and maltose (2% w/w). YNB that lacked a carbon source was used as a negative control.

Chardonnay must used in the fermentation analysis was also used to measure the growth kinetics of yeast strains tested. Moreover, synthetic media that mimicked must fermentations at pH adjusted to 3.5 and ethanol concentrations of either 5% or 10% were used (20 g/L glucose, 4 g/L tartaric acid; 0.134 g/L sodium acetate; 5 g/L glycerol; and 1.7 g/L YNB) (v/v).

Pre-cultures of *H. vineae* T02/19AF, *H. vineae* T02/05AF, *H. uvarum* AWRI1280, *H. osmophila* AWRI3579, and *S. cerevisiae* ALG804 were prepared in yeast extract peptone dextrose (YPD) media (1% yeast extract and 2% peptone, 2% glucose) via incubation for 12 h in a rotary shaker at 150 rpm and 25 °C. These pre-cultures were used to inoculate fermentations carried out in microtitler plates at a final volume of 250 μL. Inoculates producing $1 \times 10^5$ cells/mL in media were used for all strains and treatments. All conditions were tested in triplicate. Absorbance at 620 nm was measured at 30-min intervals for 48 h at 25 °C using an automatic plate reader (Tecan, Männedorf, Switzerland) and data were acquired with the Magellan software for further statistical analyses.

*2.4. Fermentation Ability in Different Carbon Sources*

The carbohydrate fermentation capacity was tested using Durham tubes immersed in media to detect gas production. Each type of medium tested was inoculated to produce a final concentration of $1 \times 10^6$ cells/mL in a final volume of 8 mL performed in triplicate. Results were visually assessed after a 48 and 96 h static incubation period at 28 °C.

*2.5. Genomic Analysis*

Genomic DNA was obtained from *H. vineae* cultures grown in a YPD medium at 30 °C using the Wizard Genomic DNA Purification Kit (Promega, NY, USA), according to the manufacturer's instructions.

The Illumina Genome Analyzer Iix platform in paired end mode was used to perform genomic sequencing as described previously [22]. Gene prediction was carried out using Augustus [30] trained

with *S. cerevisiae* gene models. Peptide predictions were then annotated using BLASTp (cutoff for e-value $1^{-10}$) against *S. cerevisiae* proteins, obtained from the *Saccharomyces* Genome Database [31].

A dendrogram was constructed using the sequences of nine genes encoding components of pathways related to glycolysis and fermentation from the *Hanseniaspora* species and *S. cerevisiae*. The genes assessed were *CDC19, FBA1, PGI1, PFK1, PFK2, HXK2, ENO1, PGK1,* and *PDC1*. *Schyzosaccharomyces pombe* was used as an external group. Neighbour joining and Kimura 2-parameter methods were carried out using the MEGA version 4 software [32,33].

## 2.6. Transcriptomic Analysis

Fermentations were performed in triplicate using chemically defined grape (CDG) must with a composition similar to that of natural grape juice, but devoid of grape precursors. Components of CDG must were defined as described in Carrau et al. [34], with modifications. Briefly, glucose and fructose were added in equimolar concentrations until a total sugar concentration of 200 g/L was reached. Vitamins and salts were added as previously described [35]. Yeast available nitrogen (YAN) content was adjusted to 100 mg N/L. Of this total, 50 mg N/L corresponded to amino acids and 50 mg N/L corresponded to diammonium phosphate (DAP) supplementation, as described previously [35]. The pH of the media was adjusted to 3.5 using HCl and a final concentration of 10 mg/L ergosterol was the only lipid provided.

Pre-cultures of *H. vineae* T02/19AF were prepared in a CDG medium and incubated 12 h in a rotary shaker at 150 rpm and 25 °C. The pre-cultures were subsequently used to inoculate fermentation reactions carried out in 250 mL Erlenmeyer flasks that were closed with cotton plugs to simulate microaerobic conditions. For all strains, fermentations were performed using 125 mL CDG and an inoculum to produce $1 \times 10^5$ cells/mL in the final medium. Static batch fermentations were conducted at 20 °C to simulate winemaking conditions.

Wine samples for transcriptomic analyses were taken during the fermentation process at day 1 (exponential growth), day 4 (end of exponential phase), and day 10 (stationary phase of fermentation). For transcriptomic studies, total RNA obtained from *H. vineae* T02/19AF isolated from three replicates sampled from three different fermentation stages (days 1, 4, and 10) were analysed independently. The nine samples were paired-end sequenced using Illumina MySeq. Trinity software was used to assemble raw reads from transcriptomic analyses and further statistical analyses were performed as specified by Giorello et al. [22].

## 2.7. Statistical Analysis

All the treatments were performed in triplicate and the statistical error was calculated as the standard deviation of all data analysed. To compare growth and fermentation kinetics, variance comparison was performed by the ANOVA test carried out with STATISTICA 7.0 software. Differences in the mean absorbance or weight loss were evaluated using the Tukey test.

## 3. Results and Discussion

### 3.1. Fermentative Capacity of H. vineae in Different Media

*Hanseniaspora* species used a limited number of carbon sources, which may have been related to the reduced competitiveness of the species throughout fermentations [27]. Regarding growth in different carbon sources (Figure 1A), growth of all the *Hanseniaspora* strains tested on both glucose and fructose had kinetics similar to that of *S. cerevisiae* ALG804. The media supplemented with sucrose was fermented by *S. cerevisiae* in a similar manner as that of media containing simple hexose. *H. uvarum* AWRI1280, however, did not grow on media containing sucrose. *H. vineae* T02/05AF, *H. vineae* T02/19AF, and *H. osmophila* AWRI3579 were able to grow on and ferment sucrose to an extent. Invertase gene (*SUC2*) is present in the genome of *H. vineae*. *SUC2* is highly expressed on day 4 of fermentation reactions, but not day 1 or 10. However, other invertase homologs were not observed in

the genomes of any other *Hanseniaspora* species except *H. osmophila* [25]. Recently, Steenwyk et al. [36] grouped *H. vineae* and *H. osmophila* within the slower-evolving linage of *Hanseniaspora*. In this branch, the *SUC2* gene is present. This is different in species of the fast evolving linage including *H. uvarum*, *H. opuntiae, H. valbyensis,* and *H. guilliermondii,* which might have lost the gene as a result of rapid mutation rates [36]. The same fact was detected with another key gene that show *Saccharomyces* wine yeast adaptations. Increased sulfite tolerance conferred by *SSU1* (Table 2) is present in *H. vineae* and *H. osmophila* and it is absent in the other *Hanseniaspora* species. The presence of *SUC2* and *SSU1* genes are indicators of adaptations to alcoholic fermentation in yeast [37].

Glycerol was not used as a unique carbon source for *H. vineae* in accordance with data reported by Albertin et al. [38]. However, Hv*GUT1* and Hv*GUT2* genes were present in the genomes of both *H. vineae* strains analysed. In addition, xylose was not used by the *H. vineae* strains as expected. A finding that was likely due to the lack of enzymes needed to carry out the xylose conversion. The group of genes were also determined to be absent in *H. guilliermondii, H. uvarum,* and *H. opuntiae* [27]. However, *H. vineae* T02/05AF and T02/19AF have the ability to grow weakly when maltose is provided as a sole carbon source, despite the fact that they were not able to ferment the sugar. The same behaviour was also observed for *H. jakobsenii* [36].

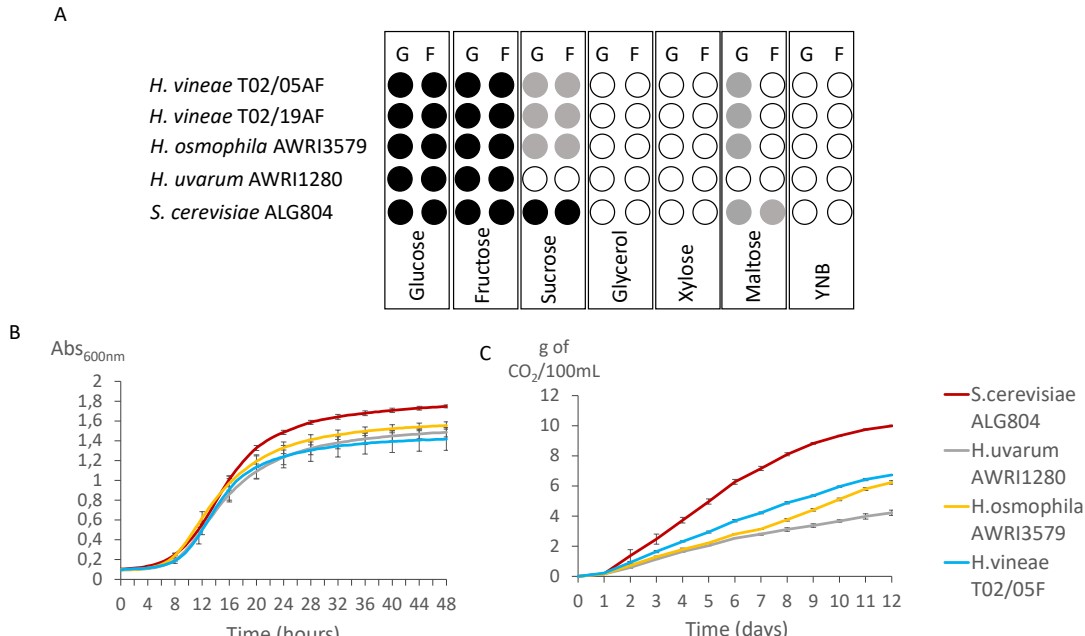

**Figure 1.** Capacity of *Hanseniaspora vineae* and *Saccharomyces cerevisiae* to grow and ferment under varied conditions. (**A**) Growth of *H. vineae* and *S. cerevisiae* (G) and the capacity of the species ferment (F) when six different carbon sources (2% w/w) were provided. Yeast nitrogen base (YNB) that lacked a carbohydrate was used as a negative control. Black filled circles indicate that full growth and fermentation were observed, grey circles indicate the moderate capacity of species to grow and ferment, and white circles indicate that the species was not able to grow or ferment. (**B**) Growth kinetics of *Hanseniaspora spp.* and *S. cerevisiae* on the Chardonnay grape juice measure as increased absorbance over a period of 48 h. (**C**) Fermentation kinetics of the three strains in the Chardonnay grape juice after 12 days are shown. Growth and fermentation experiments were performed using independent triplicate samples and error bars express standard deviation.

**Table 2.** Genes involved in sugar transport, glycolysis, and alcoholic fermentation from *S. cerevisiae* and *H. vineae*. Gene copy numbers are detailed in brackets.

|  | *S. cerevisiae* | *H. vineae* |
|---|---|---|
| Sugar transport and sensors | *HXT* (x17); *SNF3*; *RGT2*; *FPS1*; *GPR1*; *GUP1*; *GUP2*; *STL1*; *JEN1*; *ASC1*; *ASC2*; *GPA2* | *HXT* (x2); *SNF3*; *GPR1*; *GUP1*; *STL1* (x2); *JEN1*; *ASC1*; *GPA2* |
| Glycolysis | *HXK1*; *HXK2*; *PGI1*; *PFK1*; *PFK2*; *FPBA1*; *TPI1*; *TDH1*; *TDH2*; *TDH3*; *PGK1*; *GPM1*; *ENO1*; *ENO2*; *CDC19*; *PYK2* | *HXK2*; *PGI1*; *PFK1*; *PFK2*; *FPBA1*; *TPI1*; *TDH2*; *TDH3*; *PGK1*; *GPM1*; *ENO1*; *ENO2*; *CDC19* |
| Alcoholic fermentation | *PDC1*; *PDC2*, *PDC5*, *PDC6*; *ADH* (x8) | *PDC1*; *ADH* (x8) |
| Key genes of wine yeasts adaptations | *SSU1*; *CUP1* (x2); *SUC2*; *THI5*; *THI11*, *THI12*, *THI13*; *THI14*; *THI16*; *THI20*; *THI21*; *THI72*; *THI73*; *THI80*; *TPC1* | *SSU1*; *SUC2* *THI7*; *THI72*; *THI80*; *TPC1* |

As expected, *Saccharomyces* was able to grow and ferment sugars faster than *Hanseniaspora* species and significant differences between the species occurred after 16 h. The growth kinetics of the three *Hanseniaspora* strains tested were similar on grape must (Figure 1B), however fermentation kinetics of *H. vineae* and *H. osmophila* revealed that these species consume sugars significantly faster than *H. uvarum* (Figure 1C).

*3.2. Sugar Transport*

The transport of sugars into the cytosol of cells is a key step of the glycolytic pathway. *S. cerevisiae* is able to detect extracellular nutrients and make metabolic adjustments that rapidly facilitate the use of extracellular compounds [39].

Of the multiple sensors described in *S. cerevisiae*, *H. vineae* possessed the following genes Hv*SNF3*, Hv*GPA2*, Hv*GPR1,* and Hv*ASC1*, which were determined to be associated with the hexose sensing capacity of both T02/19AF and T02/05AF strains (Table 2). Sc*SNF3* encodes a low glucose sensor present in the plasma membrane that is involved in the regulation of glucose transport and also has the capacity to sense fructose and mannose in *S. cerevisiae*. Expression of the gene in *H. vineae* increases throughout fermentation (Figure 1). Sc*GPA2*, Sc*GPR1*, and Sc*ASC1* are hexose sensors that have been reported to be necessary for fermentation and are part of the "fermentome" in *S. cerevisiae*. Deletion of the genes was previously reported to induce protracted fermentation [28]. Hv*GPA2* and Hv*GPR1* have similar expression patterns throughout the fermentation process. The genes are maximally expressed on day 4 and their expression levels decrease at day 10. On the other hand, Hv*ASC1* is most highly expressed on the first day of fermentation and levels were drastically reduced both on day 4 and 10 relative to day 1 (Figure 2).

*S. cerevisiae* possesses 20 sequences putatively associated with the hexose transport [40]. *H. guilliermondii* UTAD222 possess 22 sugar transporters, and based on their DNA sequences, ten were predicted to be associated with the hexose transport, all of them were most similar to *HXT2* [27]. A comparison of sugar transporters of both sequenced strains of *H. vineae* with *S. cerevisiae* revealed that T02/05AF had two copies of the Hv*HXT6* gene and one copy of Hv*HXT1*. Sequences homologous to Sc*HXT2* were not found in the species. However, strain T02/19AF was determined to have a single copy of Hv*HXT6*. Expression levels of the gene increased after day 10 of fermentation. No sequences homologous to *HXT1* were identified. Sc*HXT1* is a low affinity hexose and pentose transmembrane transporter and is paralogous to Sc*HXT6* [41]. The Sc*HXT6* gene encodes a high affinity hexose transmembrane transporter that transports glucose, fructose, and mannose [42]. Tondoni et al. [43] revealed that in *S. cerevisiae* and *Torulaspora delbrueckii*, *HXT6* is most highly expressed throughout late stages of fermentation (Figure 2). In addition, both T02/05AF and T02/19AF strains have one sequence that is homologous to the *S. cerevisiae* Sc*HXT7* gene. Sc*HXT7* is a high-affinity glucose transporter that is very similar to Sc*HXT6* [42,44]. This gene is maximally expressed at day 4 at the end of the exponential phase of fermentation in *H. vineae*. Both *H. vineae* strains sequenced lacked polyol transporters (such as Sc*HXT13*, Sc*HXT17*, or Sc*HXT16* of *S. cerevisiae*) needed for the uptake of sorbitol and mannitol.

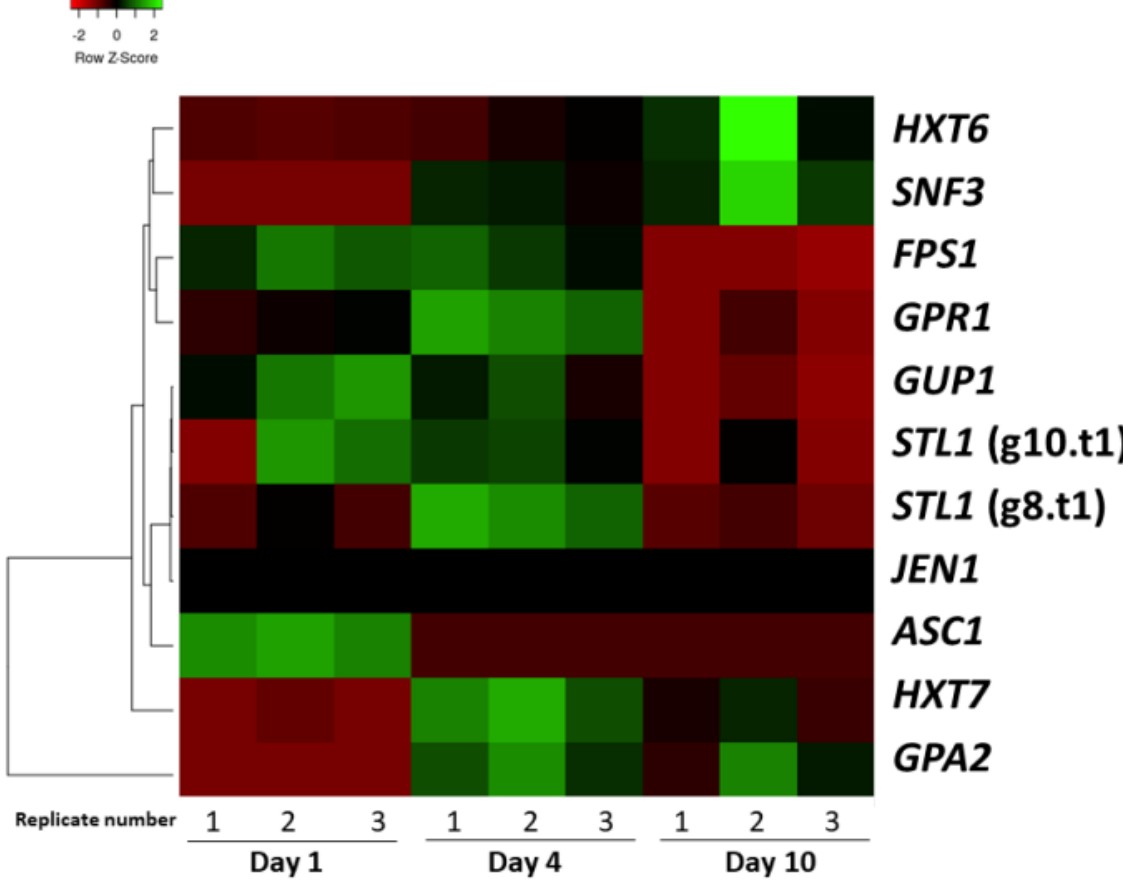

**Figure 2.** Heatmap depicting the expression levels of genes putatively involved in sugar detection and transport in *H. vineae* after 1, 4, and 10 d of fermentation. The green colour indicates elevated expression levels and the red colour indicates reduced expression levels. Data are shown in triplicate.

Other homologs of sugar transporters have been found in *H. vineae*. Three tandem copies of Hv*STL1* were identified in both T02/19AF and T02/05AF strains that shared homology with a glycerol proton symporter of the plasma membrane, which has been shown to be inactivated in response to glucose in *S. cerevisiae* [45]. Other *Hanseniaspora* species sequenced, such as *H. osmophila*, *H. opuntiae*, *H. guilliermondii*, *H. uvarum*, and *H. valbyensis* also possessed between two and four copies of the gene. According to the transcriptomic analyses, just one of the copies identified was differentially expressed throughout fermentation in *H. vineae* (Figure 2).

Hv*FPS1*, a putative plasma membrane channel involved in glycerol and xylitol movement, is present in the genome of both T02/19AF and T02/05AF. Expression of the gene is elevated near the beginning of fermentation reactions (days 1 and 4) and decreased at day 10 (Figure 1). One copy of Hv*GUP1* was present in each strain analyzed as well as Hv*JEN1*. Moreover, it was suggested that Sc*GUP1* participates in glycerol transport and Sc*JEN1* mediates the high-affinity uptake of lactate, pyruvate, and acetate so that they can be used as carbon sources in *S. cerevisiae* [46,47].

### 3.3. Glycolytic Pathway in H. vineae Strains

The first enzyme of the glycolysis pathway is a hexokinase (Figure 3). Sc*HXK2* phosphorylates glucose in the cytosol. In *S. cerevisiae*, this isoform is principally responsible for glucose activation, which is needed to initiate glycolysis when glucose is provided as a carbon source and inhibits Sc*HXK1* [41]. However, in *H. vineae*, Hv*HXK2* was the only enzyme identified with putative hexokinase activity, amino acid homology was higher compared to other species of this genus.

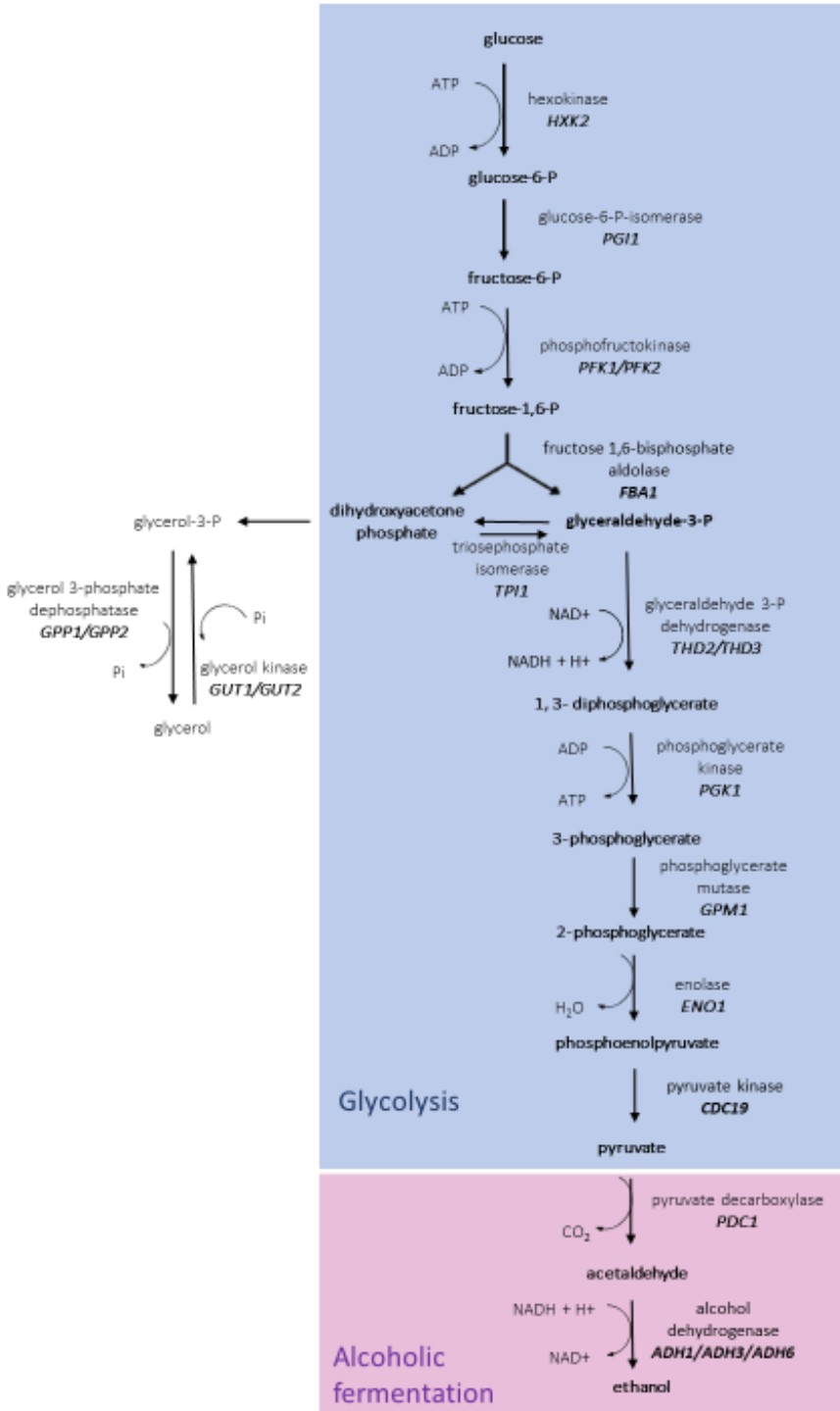

**Figure 3.** Glycolysis and alcoholic fermentation pathways in yeast. Genes putatively predicted to be involved in the catabolic pathway based on sequence data from genomic analyses of *Hanseniaspora vineae* strains are presented.

Phosphofructokinase activity was determined to be the second-most important glycolytic enzyme. The enzyme determines fermentation capacity and is indispensable for anaerobic growth. In *S. cerevisiae*, the enzyme is composed of two alpha and beta subunits that are encoded by Sc*PFK1* and Sc*PFK2*, respectively. *Hanseniaspora* strains possess sequences homologous to both Sc*PFK1* and Sc*PFK2* subunits and similar to *S. cerevisiae*, the subunits form a hetero-octameric complex [29]. Protein

sequences of both Hv*PFK1* and Hv*PFK2* were most similar to *S. cerevisiae* (76.78% and 79.24%) and *H. osmophila* (76.46% and 77.50%) relative to the other *Hanseniaspora* species assessed (Figure 4A). Phosphofructokinase only works in the forward direction and is not involved in gluconeogenesis. In fact, three activities are required for gluconeogenesis: Pyruvate carboxylase, phosphoenolpyruvate carboxykinase, and fructose-1,6-bisphosphatase. No genes encoding the key gluconeogenic enzymes have been identified in *H. vineae, H. guilliermondii, H. uvarum, H. osmophila,* or *H. valbyensis* [27]. This explains why *Hanseniaspora* species are not able to grow when non-carbohydrate precursors such as pyruvate, amino acids, or glycerol are provided as energy sources. This is different than *S. cerevisiae*, which is able to grow on a variety of carbon sources including ethanol and lactate [48].

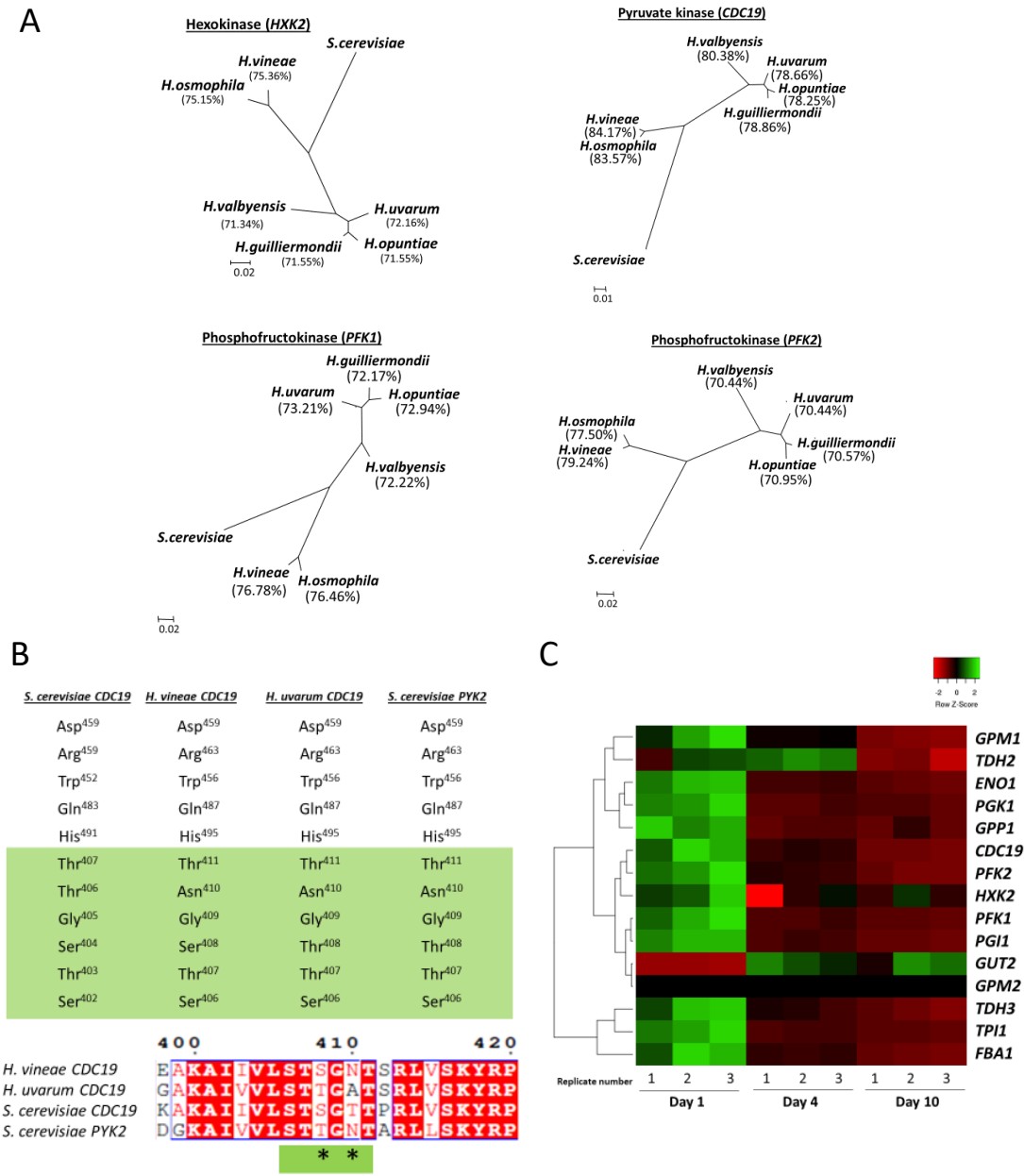

**Figure 4.** Genes involved in glycolysis. (**A**) Dendrograms showing the genetic distances between predicted amino acid sequences of enzymes involved in glycolysis from seven *Hanseniaspora* species and the *Saccharomyces cerevisiae* S288c strain. Amino acid homology was calculated for each *Hanseniaspora* strain against *S. cerevisiae*. (**B**) Amino acid sequences that correspond to the binding domains of fructose

1,6-bisphosphate inducer of pyruvate kinase in *S. cerevisiae*. Sc*CDC19* and Sc*PYK2* genes were compared with predicted sequences of *CDC19* from *H. uvarum* and *H. vineae*. Amino acids corresponding to the region that differ from *CDC19* and *PYK2* are highlighted in green and the position of residues are marked with an asterisk (*). (**C**) A heatmap describing the expression levels of genes putatively determined to be involved in glycolytic pathways of *H. vineae* 1, 4, and 10 days after the initiation of fermentation. Green and red colours indicate high and low levels of expression, respectively. Data are shown in triplicate.

Predicted amino acid sequences of phosphoglucose isomerase (*PGI1*) from *H. vineae* and *H. osmophila* were 86% similar to that of *S. cerevisiae*. Predicted *PGI1* amino acid sequences from *H. uvarum*, *H. valbyensis*, *H. guilliermondii,* and *H. opuntiae* were approximately 71% similar to *S. cerevisiae*. This tetrameric enzyme is involved in the interconversion of glucose-6-phosphate and fructose-6-phosphate. Phosphoglucose isomerase activity has also been associated with the regulation of the cell cycle and gluconeogenic events of sporulation in *S. cerevisiae* [49,50].

Two copies of the *S. cerevisiae* Sc*ENO1* gene that encodes an enolase were identified in *H. osmophila*, while only one copy was identified in other sequenced *Hanseniaspora* species. This enzyme catalyzes the conversion of 2-phosphoglycerate to phosphoenolpyruvate during glycolysis. Glyceraldehyde-3-phosphate dehydrogenase (GAPDH) is a tetramer that catalyzes the conversion of glyceraldehyde-3-phosphate to 1,3 bis-phosphoglycerate. Three unlinked genes, Sc*TDH1*, Sc*TDH2*, and Sc*TDH3*, encode related, but not identical, polypeptides that form catalytically active homotetramers with different specific glyceraldehyde 3-phosphate dehydrogenase activities in *S. cerevisiae* [51,52]. In *H. vineae*, only *TDH2* and *TDH3* homologues have been identified, and both were differentially expressed throughout fermentation (Figure 4C).

*H. vineae* strains T02/19AF and T02/05AF possess Hv*GUT1* and Hv*GUT2* genes. Both Sc*GUT1* and Sc*GUT2* are associated with glycerol kinase activities in the cytoplasm and mitochondria, respectively. Glycerol degradation is a two-step process that is mediated by *GUT1* and/or *GUT2*. Under aerobic conditions, *S. cerevisiae* is able to utilize glycerol as a sole carbon and energy source [53]. Both of the enzymes have homologs that have been identified in *H. vineae* and *H. osmophila*, other *Hanseniaspora* species such as *H. uvarum*, *H. guilliermondii*, and *H. opuntiae* lack homologous of these genes [27].

Several specific activities associated with glycolytic enzymes of *S. cerevisiae* and *H. uvarum* have high degrees of similarity, which highlights the general conservation of glycolytic pathways and the downstream reactions involved in ethanol production [29]. Pyruvate kinase is a key enzyme that catalyses an irreversible step of the glycolytic pathway. The position of the enzyme at the branchpoint between fermentation and respiration makes it a key determinant energy metabolism [54]. Recent work revealed that the pyruvate kinase activity enhanced the capacity of *S. cerevisiae* to ferment sugars versus *H. uvarum* [29]. The predicted proteins, Cdc19p, of *H. vineae* and *H. osmophila* are more homolog to the corresponding Cdc19p of *S. cerevisiae* than those of *H. uvarum* and other *Hanseniaspora* species (Figure 4A). When residues of the catalytic domain of ScCdc19p [55] are compared with those of *H. uvarum* and *H. vineae,* only one amino acid difference was identified; Asp$^{265}$ was substituted with Gly$^{269}$ in *H. uvarum* and *H. vineae* (Figure 4B). However, in the binding site of the allosteric activator, fructose 1,6-bisphosphate, two amino acid differences between *H. uvarum* and *S. cerevisiae* and one between *H. vineae* and *S. cerevisiae* were identified. The two differences identified between *H. uvarum* and *S. cerevisiae* are at the same positions (Figure 4B) as those identified in the *PYK*2 gene of *S. cerevisiae,* a paralog of *CDC*19 that is characterized by its low pyruvate kinase activity compared with the pyruvate kinase protein encoded by *CDC*19 (formerly *PYK1*) [54].

Expression levels of 13 *H. vineae* genes involved in the glycolytic pathway mainly decreased from day 1 to day 4 of fermentation and were maintained throughout the stationary phase (Figure 4C). This finding is in agreement to previous observations in *S. cerevisiae* [43]. However, levels of Hv*TDH2* expression remained high at both days 1 and 4 and decreased expression levels were observed at day 10. Additionally, expression of Hv*GUT2* peaked at days 4 and 10, and increased expression levels of the gene were not detected at day 1. Finally, Hv*GPM2* was not expressed under the conditions assessed.

*3.4. Alcoholic Fermentation in H. vineae Strains*

The pyruvate decarboxylase activity plays a key role in the alcoholic fermentation pathway. Three different pyruvate decarboxylase isozymes have been identified in the genome of *S. cerevisiae*: Sc*PDC1*, Sc*PDC5,* and Sc*PDC6*. The function of pyruvate decarboxylase is the degradation of pyruvate into acetaldehyde and carbon dioxide. The enzyme is responsible for transferring the final product of glycolysis (pyruvate) to ethanol production [56]. In *H. vineae*, no sequences homologous to Sc*PDC5* and Sc*PDC6* were found and Hv*PDC1* was the only pyruvate decarboxylase isozyme identified in the species. In *S. cerevisiae*, Sc*PDC1* was strongly expressed in fermenting cells. The enzyme is conserved among yeast, bacteria, and plants. It is regulated by glucose and ethanol concentrations and also by itself [57]. The active enzyme has a homotetrameric structure and the enzyme has two known cofactors: Thiamin diphosphate (ThDP) and $Mg^{2+}$ [58–60]. In *H. vineae*, genes involved in thiamine biosynthesis have not been identified and a similar finding was also reported in *H. guilliermondii* [27] and most other *Hanseniaspora* species [36]. It has been suggested that this may contribute to the low alcoholic fermentative capacity of *Hanseniaspora* species, the phenotype has been shown to be related to the weak pyruvate kinase activity of *H. uvarum* [29]. *S. cerevisiae* genes associated with thiamine production are upregulated in the stationary phase of growth. Oenological strains with improved expression levels of the genes have corresponding elevated rates of fermentation [61]. This phenomenon may result from vitamin depletion that occurs after the exponential phase.

Alcohol dehydrogenases, which catalyse the conversion of acetaldehyde to ethanol are key fermentative enzymes. Many alcohol dehydrogenases have been identified in *S. cerevisiae* including Sc*ADH1*, Sc*ADH2*, Sc*ADH3*, Sc*ADH4*, Sc*ADH6,* and Sc*ADH7*. Many homologues of *S. cerevisiae* alcohol dehydrogenases have been found in the *H. vineae* genome. *H. vineae* has the same number of copies of the genes as *S. cerevisiae*. Eight alcohol dehydrogenase genes are present in *H. vineae* species, compared to six in *H. osmophila,* and four in other sequenced species of *Hanseniaspora* such us *H. uvarum*, *H. guilliermondii*, *H. valbyensis,* and *H. opunt*iae. This may explain the improved adaptation of *H. vineae* to alcohol fermentation relative to other *Hanseniaspora*. It is noteworthy that of the eight Hv*ADH* sequences found in the genome of *H. vineae*, at least three Hv*ADH6* genes are encoded in tandem. Increased copies of the gene may be associated with increased fermentation capacity, indicating that the alcohol dehydrogenase activity might be a key feature of alcoholic fermentation adaptations [62]. *H. vineae* has an enhanced tolerance to ethanol (Figure 5B) versus *H. uvarum* and *H. osmophila*, which are unable to grow in media containing 10% ethanol.

*H. vineae* and *H. osmophila* genes encoding putative alcohol dehydrogenases were grouped in two main clusters that contained either *ADH1*, *ADH2* and *ADH3* or *ADH6* and *ADH7* (Figure 5A), this is in agreement with the two multigenic families reported by Giorello et al. [22]. The clusters were formed according to the clustal alignment of predicted protein sequences, however, regarding adscription by a single homology with *S. cerevisiae ADH*s in the databases [22] produced some discrepancies. Therefore, Hv*ADH6* homologs from *H. vineae* and *H. osmophila* were removed from the Hv*ADH6* and Hv*ADH7* cluster. Moreover, the Hv*ADH1* homologous sequence of *H. vineae* is grouped in the cluster of Sc*ADH6* and Sc*ADH7*.

Hv*ADH* genes display different expression patterns (Figure 5C). Two of four paralogous copies of Hv*ADH6* were not differentially expressed at the time points analysed. Expression of one copy of *ADH6* significantly declined between days 1 and 4 of fermentation. In addition, the expression of one copy of *ADH3* was elevated on day 4 relative to day 1 (Figure 5C). These behaviours are similar to those of aryl alcohol dehydrogenases that facilitate the production of increased levels of alcohol by *S. cerevisiae* [63]. Therefore, Hv*ADH*s may be important for reducing levels of fusel aldehydes by producing increased levels of alcohol in *H. vineae* [22].

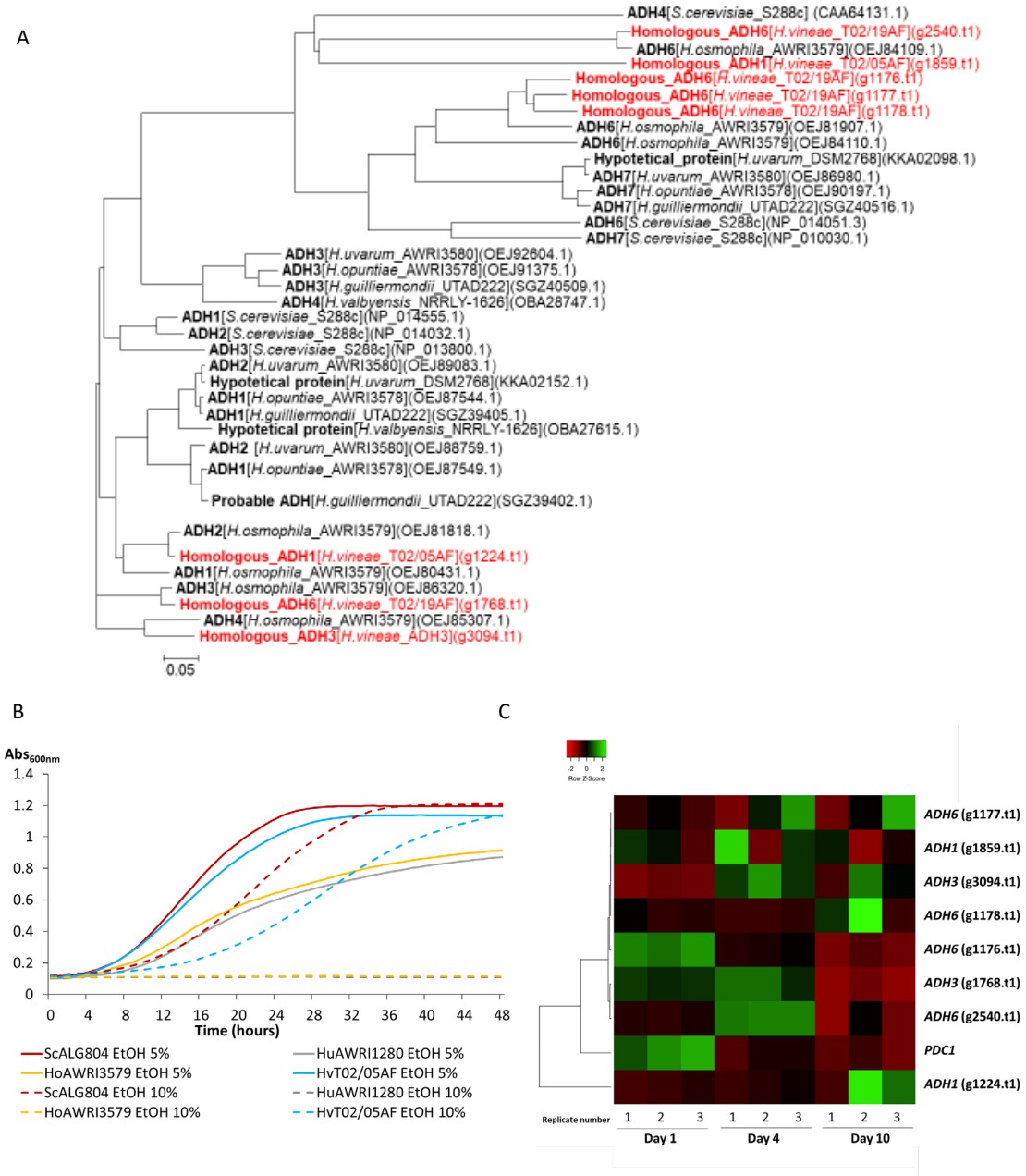

**Figure 5.** Characteristics that facilitate fermentation. (**A**) Dendrogram depicting relationships between the predicted amino acid sequences of several putative *ADH* genes. *Hanseniaspora vineae* sequences are indicated in red. (**B**) Growth of *Hanseniaspora* species and *Saccharomyces cerevisiae* in synthetic wine containing 5% of ethanol (solid line) and 10% ethanol (dotted line) for 48 h. Error bars are not shown to enhance clarity. SD < 0.05 for all samples. (**C**) A heatmap depicting expression levels of genes putatively involved in glycolytic pathways in *H. vineae* after 1, 4, and 10 days of fermentation. Green and red colours indicate high and low levels of expression, respectively. Data are shown in triplicate.

### 3.5. Hanseniaspora Genus as an Evolution Model for Alcoholic Fermentation Adaptations

The glycolytic potential of two strains of *H. vineae* were analysed using genetic, transcriptomic, and phenotypic data. Results explained the good performance of the species with respect to fermenting wine [7,21]. Findings also showed that the *H. vineae* behaviour was similar to traditional *S. cerevisiae* strains used in winemaking. Due to the outstanding capacity of *H. vineae* to produce aromatic metabolites, it was necessary to compare the capacities of the *H. vineae* strains to produce ethanol with

*S. cerevisiae.* The high degree of similarity between glycolytic and alcoholic fermentation enzymes of *H. vineae* and *H. osmophila* with *S. cerevisiae* showed that the two species should be classified as fermenters, while the remaining *Hanseniaspora* species assessed were adapted to the fruit niche and were correspondingly included in the fruit group. In our experience, *H. vineae* strains cannot be isolated from the fresh grape fruits [19]. A dendrogram of concatenated DNA sequences from seven glycolytic and fermentation genes (Figure 6) indicated the presence of two clades of *Hanseniaspora* species, similar to findings of Steenwyk et al. [36] determined using genes from the DNA repair processes present within the genus. Interestingly, the fruit and fermentation clades shown in Figure 6 were correlated with the slow and fast evolution lineages defined by these authors. Branches were in agreement with phylogenetic classifications that were based on ribosomal genes [19]. It might be interesting to use the group as an evolution model to determine the mechanism by which the fermentation group diverged separately from the fruit group [36], giving less species diversity probably due to slow evolution mechanisms. Further work will be needed to understand whether the process might be an example of domestication, as has been proposed for *S. cerevisiae* wine and beer strains [64].

Previous studies have compared the fermentation capacity of two species belonging to the fruit group: *H. guillermondi* and *H.uvarum* [27,29], and the work presented here is the first assessment of a member of the fermentation group of *Hanseniaspora*.

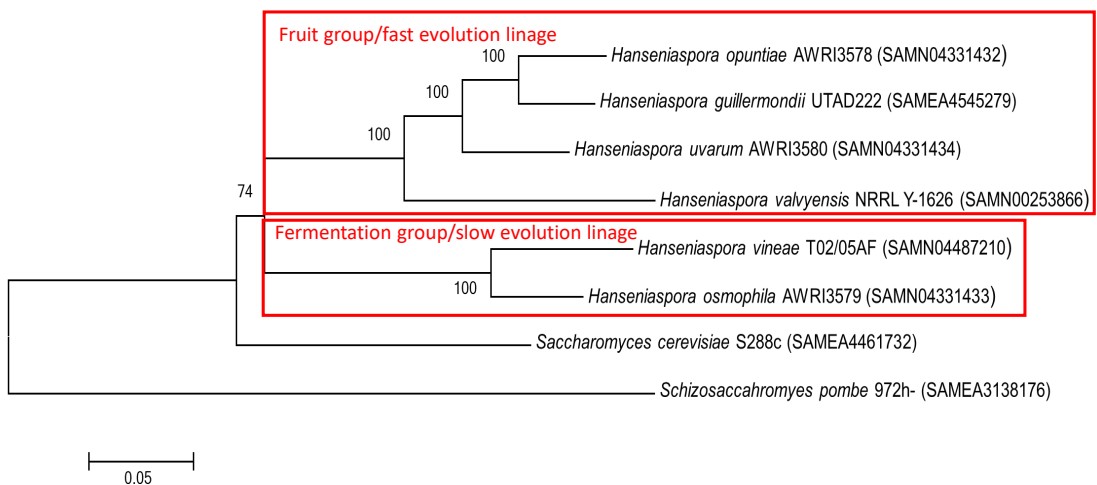

**Figure 6.** Dendrogram of seven concatenated DNA sequences from *Hanseniaspora* species constructed using the neighbour-joining method. The robustness of branching is indicated by bootstrap values (%) calculated for 1000 subsets. The entries in brackets correspond to NCBI BioSample identifiers.

## 4. Conclusions

The results suggest that *H. vineae* is clearly better adapted to the fermentation niche compared to what we named as the *Hanseniaspora* fruit clade. These results are in agreement with a separately evolution divergence between the two clades of the genus *Hanseniaspora* as was proposed previously. Phenotypic behavior of *H. vineae* growth, ethanol tolerance, and fermentation kinetics are in agreement with the genetic and transcriptomic data provided. The results obtained demonstrate that *H. vineae* and a genetically closely related species, *H. osmophila*, behave similarly. Homologies of glycolytic and alcoholic fermentation enzyme sequences of both species were compared to *S. cerevisiae*, and the similarities observed allowed the differentiation of *H. uvarum* from *H. osmophila* and *H. vineae*. High sequence homology in these latter two species was observed for key genes involved in glycolysis such as *HXK2*, which encodes hexokinase, *PFK1/PFK2* subunits of phosphofructokinase, and *CDC19* that encodes pyruvate kinase. This homology could explain the improved fermentative performance observed for *H. vineae* compared with other *Hanseniaspora* species. The elevated number of copies of *ADH* genes in *H. vineae* might be associated with increased ethanol tolerance in the species. The presence of active

genes typically related to wine fermentation capacities in *H. vineae* and *H. osmophila* such as sulfite tolerance (*SSU1*) and sucrose hydrolyzing invertase (*SUC2*) differentiate both species from the other sequenced species of the genus. Taken together, findings reported here support the characterization of the *Hanseniaspora* genus into two different groups that are adapted to two different niches, fruit and juice fermentation. These results have contributed to the improved characterization of the genus and furthermore might support the importance of it as a model for further studies related to the genetic and evolutionary phenomena of yeast domestication processes.

**Author Contributions:** M.J.V., E.B., E.D., and F.C. conceived the study and its design; M.J.V. and F.C. wrote the manuscript; M.J.V. performed laboratory experiments and data analysis; E.B. carried out statistical analysis. All authors read and approved the manuscript.

**Funding:** This research was funded by Agencia Nacional de Investigación e Innovación (ANII), Application of *Hanseniaspora vineae* Project ALI_2_2019_1_155314 with Lage y Cia-Lallemand, Uruguay.

**Acknowledgments:** We wish to thank our Universidad de la Republica for basic support of this work: CSIC Group Project 802 and Facultad de Quimica, Uruguay.

**Conflicts of Interest:** The authors declare no conflict of interest.

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
