# Peer review of "Comparison of the Glycolytic and Alcoholic Fermentation Pathways of Hanseniaspora vineae with Saccharomyces cerevisiae Wine Yeasts"

_fermentation, doi:10.3390/fermentation6030078_

Round 1

Reviewer 1 Report

This study is a comparison of fermentation pathways of Hanseniaspora vineae and Saccharomyces cerevisiae, and shows differences which can help explain their roles in winemaking. It is a clear and well presented study.

What was the composition of the Chardonnay grape juice used in the study?

Reference 50: Hochmann S is incorrect - should be Hohmann S

line 94:  insert 'have' ... which might have lost the gene ...

line 118-119: should be Figure 1B and 1C?

line 561: should be Table 1, not Table 2

Author Response

This study is a comparison of fermentation pathways of Hanseniaspora vineae and Saccharomyces cerevisiae, and shows differences which can help explain their roles in winemaking. It is a clear and well presented study.  THANKS

What was the composition of the Chardonnay grape juice used in the study?

IT WAS ADDED IN THE TRACKING VERSION WHERE YOU CAN FOUND THE CHANGES, WE ADD YAN, SUGARS AND Ph DATA.

Reference 50: Hochmann S is incorrect - should be Hohmann S   DONE

line 94:  insert 'have' ... which might have lost the gene ...DONE

line 118-119: should be Figure 1B and 1C?   CORRECTED.

line 561: should be Table 1, not Table 2   CORRECTED

Reviewer 2 Report

The introduction must clearly state the aim of the study.

The conclusion is not clearly written. It is necessary to specifically describe what are the differences between the groups species of Hanseniaspora genus revealed in the study.

I recommend major revisions.

Author Response

The introduction must clearly state the aim of the study.

WE ADDED NOW IN THE TRACKING VERSION YOU CAN SEE THE AIM.  WE ADD A NEW REFERENCE.

The conclusion is not clearly written. It is necessary to specifically describe what are the differences between the groups species of Hanseniaspora genus revealed in the study.

WE MADE A CONCLUSION NOW MORE CLEAR, AND WE THINK IT WAS IMPROVED WITH DIFFERENCES RESUMED. ENGLISH WAS REVISED ALSO.